# From "Whither" to "Whence": A Decolonial Reading of Malabou

Rachel Cicoria

Department of Philosophy, Texas A&M University, College Station, TX 77843-4237, USA; rmcicoria@tamu.edu

**Abstract:** A turn from the "whither" to the "whence" of anarchism is at stake in Catherine Malabou's interpretation of Latin American decolonial theory. This is a turn from a materialist philosophy that seeks to open the space of anarchism within the modern state toward one that discerns anarchism as already operative in the modern state given the social implications of colonial legacies. In tracing this turn, I propose a development of Malabou's work insofar as I put her in dialogue with María Lugones, who is much closer to Malabou than the more canonical decolonial figures she actively engages, especially in view of anarchism as a form of social–political plasticity. Understanding Lugones' critique of earlier iterations of decolonial theory helps make explicit an immanent anarchic resistance to domination as an explosive inhabitation of everyday loci of tension.

**Keywords:** Catherine Malabou; María Lugones; new materialism; plasticity; anarchist theories; decolonial theory; deconstruction; decolonial feminism; agency; resistance





## 1. Introduction

A turn from the "whither" to the "whence" of anarchism is at stake in Catherine Malabou's discussion of the confrontation between deconstruction and decolonial modalities of critique. This is a turn from materialist philosophies that, looking to plasticity as a model of resistance to hegemony [1] (p. 4), seek to deconstructively open a space of anarchism within the social conditions of the "teleocratic" modern state toward one that discerns anarchism as already operative in the modern state given the social implications of colonial legacies. In tracing the turn from "whither" to "whence", I propose a development of Malabou's work by putting her in dialogue with María Lugones, whose decolonial feminism is much closer to Malabou than the more canonical decolonial figures she actively engages, especially in view of anarchism as immanent resistance to domination. Like Malabou, Lugones could be said to pursue an "anarchic ontology" in the sense that her approach to (post)colonial social contexts, as dynamic and heterogeneous fields constituted by divergent modern and non-modern social logics, reflects "a structure of difference that forms itself and its meaning from an encounter that itself has no predetermined, anterior 'meaning'" [1] (p. 13). In particular, understanding Lugones' critique of earlier iterations of decolonial theory (mobilized by figures like Aníbal Quijano) helps make explicit an immanent anarchic resistance to domination as an explosive inhabitation of everyday loci of tension that is, moreover, a "remembering without having memorized".

In this respect, I utilize Malabou's concept of plasticity not only as "a resource for an active politics of resistance to our current political forms and for the creation of new alternative forms" [2] (p. 9). Following Ian James, who points out that plasticity is not only about the material mutability of form but also about the capacity of forms to be exchanged or transformed in the passage from one regime to another, including between domains of thought [2] (p. 2), I also take up plasticity as a way of approaching and facilitating a relation between "regimes" of anarchist and decolonial theories. In my view, this is an important intervention given that connections between anarchist and decolonial traditions of thought and practice remain underexplored. It is also significant in terms of the under-analyzed

role that decolonial theory plays in Malabou's own work. Finally, this paper contributes to decolonial scholarship by delimiting, in reference to Malabou's discussion of Latin American thinkers, an emerging trajectory within decolonial theory that could be said to take up social–political life in its plasticity as an ongoing site of resistance. This gestures to possibilities of further dialogue between decolonial theories, anarchist theories, and new materialisms, especially with respect to the concept of plasticity.

## 2. Plasticity

According to Catherine Malabou, the concept of plasticity (which is itself "plastic") pertains to the capacity to receive, give, and annihilate form [3] (p. 5). It applies not only to the brain but also to life more broadly. Plasticity is a material transformative potency. It is simultaneously generative in that it concerns "the quasi-infinite possibility of changes of structure authorized by the living structure itself" [4] (p. 206) and destructive in that this immanent wealth of possibility also poses the threat of explosion [3] (p. 71). This contradictory and contingent "double movement" of emergence–disappearance [3] (p. 70) does not occur continuously. Specifically, it does not unfold as the preservation of identity or the pursuit of harmony [5] (p. 11). Instead, leaving some traces while effacing others [3] (p. 74), plasticity transpires through unpredictable and eruptive transformations out of mutual conflict [3] (p. 72). With plasticity, Malabou pursues an analysis of a process, rather than a substance metaphysics [1], and counters dominant dialectical materialisms as she develops an anarchic philosophical materialism of contingency and chance: one that "doesn't presuppose any telos, reason, or cause" [4] (p. 205). In this respect, she understands the immanent dynamic of material formation (including emergent social organizations) as a non-localizable and non-intentional search for equilibrium [4] (pp. 207–208) arising from within "a sort of blank space that is the highly contradictory meeting point of nature and history" [3] (p. 72).

Malabou takes seriously the explosive dimension of plasticity, or what she calls "destructive plasticity": a plasticity that, marking the "extreme limit of plasticity in its negative form" [8] (p. 115), "does not repair. . .that cuts the thread of life in two or more segments that no longer meet" [5] (p. 6) and so annihilates equilibrium as it gives form to a general identity. Yet, destructive plasticity is not simply negative: "It is also metamorphic" [8] (p. 116). In other words, "it also forms something new, even if this something or someone is so radically different as to make recognition impossible" [8] (p. 115). In the turn from whither to whence, I consider how, by describing a destructibility already embedded within life's metamorphic potential, plasticity sheds light on the social as harboring within it a cataclysmic, but resistant, metamorphic potency [2]. This is informed by Malabou's pursuit of a radical mode of (trans)formation: one that, occurring in/with/through explosion, does not "serve the neatness and power of realized form" [5] (p. 5) and "knows no salvation or redemption and is there for no one, especially not for the self" [5] (p. 12).

At the extremes of determination and freedom [3] (p. 17), Malabou locates within life's plasticity an immanent "power to style" [3] (p. 15) that is simultaneously an agency of creation, reception, and disobedience [3] (p. 6). This agency, which cannot be appropriated by the self as its "own" [7] (p. 117), is the capacity for "transdifferentiation": the "possibility of *displacing* or *transforming* the mark or the imprint, of changing determination in some way" [3] (p. 16) [3]. Drawn from examples of stem cell plasticity, specifically their "pluripotent" capacity to change into cells of other tissues [3] (p. 16), transdifferentiation is an anarchic "form of regeneration and freedom" [8] (p. 116): a non-localizable mode of agency that, with affective indifference to explosion, inhabits embodied reservoirs of potency toward unforeseeable and irredeemable possibilities of metamorphosis. In this respect, I note that transdifferentiation resonates with Malabou's description of the logic of "epigenesis" [4].

Literally meaning "above" or "over" genesis [7] (p. 120), epigenesis more broadly pertains to "theories of coming-into-being, of the mutability of structure and organization, attesting to the inscription of contingency and temporality in being and calling attention

to the various processes through which forms emerge, endure, and are modified over time" [7] (p. 110). As Alexander Miller explains, epigenesis also identifies a political agency that emerges "at the point of convergence of reflexivity and [biological] plasticity" [7] (p. 116). In this respect, as James notes, formation occurs as the "reciprocal donation of form" [2] (p. 4). "It is here", Miller writes, "at this juncture, that the subject of epigenesis becomes aware of her own plasticity, of the reserve of malleability that inhabits her very corporeality, and thus of her capacity, indeed her obligation, to take responsibility for her own becoming" [7] (p. 117).

I find that these two senses of epigenesis—theoretical and practical—come together in Malabou's geological elucidation of the term (specifically the prefix *epi*). She writes:

> In geology, the 'epicenter' is the point of projection of the 'hypocenter,' the underground site where an upheaval emerges on the surface of the Earth. The hypocenter is the underground focus of an earthquake, while the epicenter is its surface event... The work of determining the position of the epicenter, the place where the destruction is greatest, is called 'localization' [7] (p. 120).

Drawing from this a "logic of epigenesis", Malabou argues that theories (and practices, I would add) of epigenesis [5] "all work at the surface, that is, they never refer to a hidden ground. They convey a founding *at the point of contact* and not at the root or a point of focus" [7] (p. 121). This is not to say that the logic of epigenesis is superficial. Instead, it indicates a localized effect, a "working where *it occurs*, at the contact point between underground and ground" [7] (p. 121). The challenge, then, is to "*locate* the epicenter and remain at the impact point" [7] (p. 121).

Building on engagements with the resistant implications of plasticity [6], but incorporating a decolonial attentiveness to the specificity of subaltern fractured loci, I explore an anarchic, transdifferentiating agency that, informed by the logic of epigenesis, arises when the "ground" begins to shake. In/with/through attunement to localized social–political tremors and their ongoing reverberations, this seismic resistance is a "changing of difference" that transpires as a "transindividual" process of becoming [7] (p. 117). In this respect, it is an authorship without author, an agency that "does not reflect itself, does not live its own transformation, does not subjectivize its change" [5] (p. 11) [7]. It thus embodies an anarchic awareness that freedom only takes shape in the explosiveness of life; it lives the knowledge that "if we didn't explode at each transition, if we didn't destroy ourselves a bit, we could not live" [8] (p. 116).

### 3. Anarchism and the Repression of Social–Political Plasticity

As suggested by Arianne Conty—"by providing 'guidance without chains'" [12] (p. 120)—anarchy appears to be the best political expression of Malabou's concept of plasticity and the best candidate for reconstituting community "outside the bounds of nation-state exclusions and political determinisms of all sorts" [12] (p. 120) [8]. Beyond anti-statism, Malabou notes that "the core trend of all anarchist movements, regardless of their diversity, pertains to the radical and uncompromised rejection of domination" [13] (p. 216) [9]. She describes domination as an illegitimate form of authority and totalizing mode of power in which "one person is constantly subordinated to and by another and becomes a prisoner of such a situation" [13] (p. 217). Domination thus identifies relations of power that are unambiguously abusive in that they seek to render their own subversion impossible.

Classical anarchism tends to regard power and resistance as "two clearly opposed entities" and thus determines power "as a substantial unity of forces commanding from above and to be resisted from the outside" [13] (p. 224). Malabou instead turns to non-classical anarchist thinkers who understand power and resistance as a contingent, co-constitutive entanglement that sustains "a multiple, fragmented, and creative force" [13] (p. 224), which is resonant with plasticity. From this perspective, domination is not exhaustive in that it "possesses at its core an internal line of fracture, a crack that allows for its self-subversion" [13] (pp. 217–218) [10]. As the expression of social–political plasticity, anarchism

is a non-teleological mode of resistance that can arise immanently within social–political formations governed by material logics of teleology and necessity, or what Reiner Schürmann refers to as "teleocracy".

Although Malabou does not utilize the term in her discussion of the "whither" of materialism (as discussed below), I find it helpful for understanding her view of the social. Coined by Schürmann to characterize domination as it is constituted by an assumed metaphysical "solidarity between *arkhé* and *telos*" [13] (p. 218), teleocracy "secures the order of things from arbitrariness and chaos by imposing a normative matrix upon being" [13] (p. 219). This sheds light on the modern nation state as a political model of domination that presupposes predefined social criteria as the necessary grounds of (human) politics [4] (p. 208). Within the contemporary nation state, and toward the reproduction of the social homogeneity that it requires, social formation thus occurs as an "expected or agreed-upon process" of normative conformity [4] (p. 207, 211).

In such contexts, I draw from Malabou that immanent resistance would arise from an internal fracture, or "political void", that is informed by non-teleological logics and, hence, conditioned by an "absence of meaning, telos, predetermination" [4] (p. 209). In this, the political void is a plastic locus of originary dispossession, or an anarchic zero-point, within teleocratic orders of domination: one that exceeds dominant social logics of identity, reproduction, and conformity [4] (p. 212). It is from this properly anonymous and heterogeneous locus of agency that non-hegemonic social and political forms can subversively crystallize—"singular, unpredictable, unseen, regenerating" [4] (p. 212). For Malabou, such resistance would occur as the political void is inhabited by anonymous people "without qualities, without privilege, without legacies, without tradition. People of nothing, people of valour" [4] (p. 212). In this respect, she is informed by Althusser's discussion of the "nameless man" (as an interpretation of Machiavelli's *Prince*), who begins "from nothing, from such a nameless place, from such a non-teleological formation of forms" [4] (p. 209).

For Malabou, the nameless man exemplifies an individual political expression of plasticity [4] (p. 208). This is because the nameless man is internally governed by the material tensions and variations of plastic balancing acts and not by dominant social logics or discourses. Hence, "form will emerge out of the encounter between fortune—that is, contingency—and the prince's virtue—that is, his ability to select the best possibilities that fortune offers, yet a selection made with no intention to do so" [4] (p. 209). The nameless man's virtue, in my view, is the epigenetic capacity for social–political transdifferentiation as this arises through an anarchic "power to style" (as discussed above). As a penchant for "exploding from time to time" [3] (p. 79), it is a non-reflective and non-subject-centered mode of sensuous participation in the equilibrating dynamics of material plasticity that entails becoming "sensitive to the validity and viability of differences" as they unpredictably erupt, crystallizing from out of nothing [4] (p. 212).

Yet, Malabou's account of immanent resistance remains hypothetical, especially in the case of the modern state. This, I find, is because Malabou sees contemporary society as a "teleocratic" political order of domination conditioned by the thorough exclusion of anything "impotent, chaotic, anarchic" [13] (p. 219) such that "the plastic condition is menaced or even non-existent" [4] (p. 208). For instance, she laments that the alternative logics needed for the anarchic constitution of the political void are "doomed to be repressed by teleology, anteriority of meaning, presuppositions, predeterminations" [4] (p. 207). Within the modern teleocratic state then, the kind of virtue needed for immanent resistance is normatively smothered and, as Malabou claims, "there seems to be no void" [4] (p. 210). Based on the assumed exhaustiveness of teleocratic oppression, Malabou argues that "opening the unassignable place in a global world, where [11] every place is assigned, has become the most urgent ethical and political task" [4] (p. 214). She writes:

> The determination of this void of nothingness, this point of possibility that opens all promise of justice, equality, legitimacy, cannot be presupposed and cannot be as blindly and automatically regulated as in nature either. It has to be made

possible. This is the philosophical task that appears with the end of the repressed materialism [4] (p. 212).

Against dominant logics of teleology and necessity, she explains that this "task"—or the "whither" of materialism—requires clearing "a point of void, nothingness and dispossession at the heart of the most important philosophical trends, which define themselves as materialisms" [4] (p. 213). As I see it, then, Malabou's determination of a materialist "whither" pertains to the need to create the conditions for the possibility of anarchism in modern social contexts where plasticity is repressed.

## 4. Malabou and Decolonial Theory

### 4.1. Two "Outsides"

Rather than continuing to discuss the "whither" of materialism, namely, the task of opening an anarchic political void, I pursue an alternative task based on Malabou's own writings. I call this task the "whence" of materialism. While "whither" refers to the need to clear the space for anarchism in the context of the repression of plasticity, "whence" refers to the inexhaustibility of plasticity, to its always being there beyond domination, and to the need for a different philosophical modality that is able to discern and engage it. The difference between the "whither" and the "whence", I argue, is implicit but not pursued in Malabou's analysis of contemporary critical philosophies and literature from Latin America in "Philosophy and the Outside: Foucault and Decolonial Thinking" [15].

In this text, Malabou explores the confrontation between two "outsides" of philosophy, one Western/deconstructive (through Foucault) and the other decolonial (through Enrique Dussel, Ramón Grosfoguel, among others). In this process, I find that she sheds light on philosophical resources for understanding possibilities of immanent resistance to domination. I thus draw from her discussion two ways of understanding the concept of political void as a site of social–political exteriority: one informed by the tradition of deconstruction (as the "outside" of the traditional concept of outside) and the other by decoloniality (as the "outside of the outside of the outside") [15] (p. 179). I suggest that the deconstructive "outside" aligns with the "whither" of materialism and the decolonial "outside" aligns with the "whence".

The former concept of "outside" comes from the inside of Western philosophy and, in the form of deconstructive critique, pertains to a space outside of philosophy as construed by the Cartesian tradition and its influence on modern forms of philosophy/politics. In this respect, Malabou draws from Foucault's analysis of modern literature, which for him is "a specific kind of writing that situates itself in a space where truth and falsity are deactivated and have lost their meaning" [15] (p. 179). From this, Foucault derives "I speak" as an immanent critique, or internally destabilizing force, of the Cartesian *ego cogito* of "I think, therefore I am". While "I think"—which leads to "the indubitable certainty of the 'I' and its existence" [15] (p. 181)—implicitly presupposes a philosophical domain of truth as this is established by (and as) a supporting discourse [15] (p. 180), for Foucault, "I speak" is an enunciative gesture that "distances, disperses, effaces that existence and lets only its empty emplacement appear' (Foucault 1987: 13)" [15] (p. 181). This is because "I speak" is not a truth statement and so "cannot refer to the pre-eminence or pre-existence of language, because language as such does not precede the act of speaking" [15] (p. 180). Denying the kind of discursively constituted reflexivity that appears to substantiate the *ego cogito*, "I speak" shows the speaking subject to be without ideal or essential grounds: "Behind the 'I speak', there is only an 'it speaks', an anonymous murmur ([Foucault] 1987: 11)" [15] (p. 180). In other words, beyond antecedent discursive configurations of truth and untruth and the modern form of subjectivity that they support, there is only an "anonymous ocean of language" [15] (p. 181), "a 'desert', 'an unfolding of pure exteriority'" [15] (p. 180). In this way, "I speak" constitutes an experience of the "outside" of philosophy: that is, an experience of the non-antecedence of a language of truth.

In distinction from this Western concept of the outside of philosophy, Malabou draws from Latin-American thinkers to elucidate a decolonial notion of the outside, which

"appears as the outside of the Western philosophical attempt at producing its own outside" [15] (p. 179). Specifically, she turns to Dussel who, by attending to the colonial conditioning of Descartes' philosophical formulations, assimilates the *ego cogito* "with the imperial *ego conquiro* of 'I conquer, therefore I am'" [15] (p. 181) [12]. Against a mode of thinking that presupposes inhabitation of an imperial geopolitical subject position, Dussel articulates a decolonial mode of critical thought that arises from "philosophers of the periphery" who, by inhabiting subaltern positionalities, are "'the ones who hope because they are always outside, these are the ones who have a clear mind for pondering reality... ([Dussel] 1985: 4)" [15] (p. 181) [13].

Like Western/deconstructive critique, Latin-American/decolonial critique seeks an outside of the Cartesian *ego cogito* both as an epistemological assumption and as a grounding subject formation that enables modern socialities and modes of domination. Malabou thus points out that both deconstruction and decoloniality can be seen as challenging Western philosophical imperialism in reference to sovereignty and the vision of language produced by "I think": "'the illusion of an autonomous discourse' (Blanchot 1987: 74, 80). An illusion that has for so long justified 'reason, exclusion, repression' of the outside (Blanchot 1987: 80)" [15] (p. 181). In this respect, she also notes that it could perhaps be said that both forms of critique "open" a subaltern location and, in this, design "the limits of the hegemony of the West" and engage "subaltern concepts repressed or ignored by the traditional philosophical heritage" [15] (p. 183). Moreover, insofar as they each pursue an outside that "undermines the tradition from within" [15] (p. 182), both deconstruction and decoloniality take the form of immanent critique. This is to say, they regard the "outside" of philosophy not, as Dussel might say, as "an utterly other exteriority, a sacred or divine dimension of language" [15] (p. 182), but rather, as Foucault explains, as operating in "another space than that of the usual duality between the inside and the outside" [15] (p. 182). Malabou therefore suggests that "in many respects, and at many levels, the vocabulary of critique, deconstruction and decolonisation, or rather decoloniality, seems to coincide, as it points at holes, even if tiny and imperceptible, in all systems in general" [15] (p. 182).

*4.2. Enunciative Legitimacy*

Despite these comparisons, Malabou ultimately argues that deconstruction and decoloniality do not "speak the same language" [15] (p. 182): "such a proximity between Western and decolonial thinkers is itself full of 'cracks and fractures'. The two outsides remain foreign to each other" [15] (p. 183). For her, this distinction hinges on an antecedent legitimacy tacitly assumed by deconstructive critique. I suggest that, in this way, Malabou implicitly puts deconstruction—specifically the "whence" of materialism as an opening of the political void and creation of the plastic condition—in an ambivalent relation to the possibility of anarchism.

Even though tradition as an antecedent discursive ground for truth is effaced by the gesture "I speak", Malabou notes that deconstructive critique continues to rely on tradition to garner legitimacy from it. She writes:

> However critical of the French colonial system that shapes the French language and its 'I speak', however critical and deconstructive of traditional metaphysics, it is still the case that Foucault, Derrida and all other contemporary European continental philosophers belong to a tradition that is theirs, their own. Their deconstructive gestures encounter no issues of legitimacy [15] (p. 183).

The political void (which takes shape without the support of tradition [4] (p. 212)) thus cannot be the emergent site for deconstructive critique, since the latter's horizon is constituted by a prior belonging to a single, and dominant, tradition of culture and thought: Greek–German–French philosophy [15] (p. 184), whose capacity as a grantor of philosophical legitimacy is sustained by colonial histories and contexts.

Echoing the discussion of the *ego conquiro*, this implies that the Cartesian *ego cogito*—participating in its enunciation through what has become a dominant cultural and

philosophical tradition—is, in fact, derivative of a European subject formation that denies the legitimacy of other sites of philosophical enunciation. Deconstruction is not exempt from this insofar as it continues to be sustained by and remains in possession of the "mother tongue" of European philosophy: "Foucault still speaks the language of truth when he criticizes the language of truth, because his concept of the anonymity of language is fundamentally Western and thus remains attached to the logos" [15] (p. 183). This is to say, although the "I speak" deconstructs the determinacy of the *ego cogito*, it does not interrogate the geopolitical and social conditionings of its own legitimacy and therefore remains bound by an unchallenged specific enunciative form of intelligible antecedence. Deconstruction thus continues to speak truths because it retains, as the condition of its enunciation, the very structure of antecedence that the "I speak" is supposed to renegue.

In this way, a decolonial analysis of an antecedently granted philosophical legitimacy shows that a form of antecedent intelligibility is still operative in the syntax of deconstruction, one that commits it implicitly to teleocratic social–political formations that are correspondingly structured, such as the modern nation state. In this way, deconstruction can be seen as a manifestation of the social formations operative through the colonial lineages of the modern state and not as an anarchic inhabitation of a political void. The point, then, is not that deconstruction cannot be critical of colonial domination but that the enunciative locus of deconstructive critique assumes the legitimacy colonial domination grants to philosophical discourse by retaining a structure of antecedent intelligibility. I thus find that deconstruction is not well-equipped to break away from the kind of teleocratic socialities that Malabou wants to resist and, moreover, is not well-equipped to see the possibility of an anarchic, plastic sociality that does not reflect the syntax of antecedence at the core of mainstream Western philosophy.

It may seem at first that this critique is about socio-historical conditions that are ultimately external to the content and purview of philosophical approaches, including deconstruction. Malabou, however, contests this. Extending her argument, I point out that because deconstruction (even the "I speak") is shaped by an antecedent intelligibility granted by the very kinds of social teleocratic formations that it seeks to undermine, the political void and anarchic plasticity appear to be inaccessible phenomena for it. More precisely, a politics of anarchic plasticity necessarily entails modes of critique and theoretical analysis that are *not* legitimized antecedently, showing a modality of enunciation that eludes the philosophical registers of deconstruction. I suggest that deconstruction, given its conceptual, syntactical, and phenomenological constrictions, can get only as far as proposing the task of opening an anarchic political void—the "whither" of materialism—but is unable to discern anarchism as already taking place. In this sense, what I am proposing here as a turn from "whither" to "whence" is anticipated by Malabou's distinct form of decolonial critique of deconstruction.

To recognize such a possibility, that of resistance to hegemony as already ongoing, one needs to engage an anarchic philosophical thought that is not antecedently structured, including not being legitimated by geopolitical contexts and philosophical traditions in advance; one that, as discussed above, *begins* from "a point of void, nothingness and dispossession" [4] (p. 213). Going beyond deconstruction, I suggest that Malabou's interpretation of decolonial critique is set to discern this kind of anarchic resistance. Yet this discernment does not point to the philosophical task of "whither", to opening the political void; for, as deconstruction implies, such a task is itself made possible by antecedent social-cultural–political support. What is pointed to is rather a "whence": to decolonial enunciation as it is already arising immanently from sites of "illegitimacy" present within, yet in excess of, teleocratic social formations. Political voids, from this perspective, do not need to be opened through a deconstructive critical gesture. The plastic condition does not need to be created or made possible. Resonant with the logic of epigenesis, the task instead appears as one of "seeing" ongoing forms of anarchic resistance to colonial domination, of locating the epicenters of social–political upheaval and remaining at the impact points [7] (p. 121).

*4.3. Definitive Exile*

Although the antecedent legitimacy afforded to deconstructive thinkers by their embeddedness in a dominant tradition does not exempt them from experiencing social rejection, Malabou notes that "however painful and unjust such an exile can be, it will never coincide with a *definitive exile* from the inside of the philosophical European canon, with its good and pure origin, its Greek-German-French nobility" [15] (p. 183, my emphasis). To this end, she argues that, for the exiled deconstructive philosopher "European philosophy always remains a mother tongue... This point is missing in Foucault's beautiful analysis in *The Thought from Outside*. When the 'I speak' turns back on itself, it is not forced to silence; the outside of language is still a language" [15] (p. 184). Malabou thus, in my view, rightfully moves to differentiate between deconstructive and decolonial loci of enunciation as distinctive geopolitical sites of critique.

While a deconstructive locus reflects the dominant European positionality of "I conquer", a decolonial locus reflects the subaltern positionalities of the "conquered". In this respect, Malabou points out that the situation from which deconstruction arises:

> ... is very different, incompatible maybe, with that of countries that certainly have their own philosophers and their own philosophical traditions, but whose philosophers and traditions are not considered canonical, or are little known in Europe, even if this situation is slightly and fortunately changing today. It is as if these non-European philosophers and traditions remained outside, outside the outside, and, if we want to play a bit more, outside the outside of the outside. The colonisation of literature [15] (p. 184).

Situated within colonial contexts where "philosophical language was initially brought from outside, that is, from Europe" [15] (p. 184), in my view, decolonial loci can be seen as political voids; as illegitimate sites of enunciation constituted by "definitive exile", by originary dispossession, rather than by prior belongingness to a globally dominant tradition. This suggests a mode of critique that does not culminate in anonymity (through a deconstructive effacement of the *ego*, for instance) but instead arises from anonymity as an antecedent condition produced by colonial oppression.

I return to the image of the earthquake, evoked by Malabou to elucidate the logic of epigenesis, to shed further light on the difference between deconstructive and decolonial loci. Through deliberate critical interventions, deconstruction seeks to destabilize the Earth—here understood as the Western philosophical tradition—as a securely inhabited prior ground. Indeed, it is the very antecedent stability of this "ground" (as a condition sustained through colonially inflected social and institutional mechanisms of legitimation) that makes possible deconstruction, specifically in the form of critical enunciations arising from dominant subject positionalities. In my view, this is the kind of task called for by the "whither" of materialism. Decolonial critique, on the other hand, is not supported by the prior stability of the "Earth" but rather presupposes the Earth's ongoing tremoring as an antecedent condition. I thus delimit decoloniality as a form of critique that arises in the midst of an earthquake (understood broadly in order to include lived experiences of eruptions conditioned by social, political, economic, discursive, environmental, etc. forces), as opposed to deconstruction which seeks to produce a theoretical earthquake as a consequence of critique.

I also note that from a deconstructive perspective, there is a steadiness to the defining contrast between stability ("I think") and instability ("I speak" as the dispossession a prior stable state)—an antecedent determinacy that, as the above discussion suggests, is conditioned by colonial domination. However, no such orienting distinction, which frames instability in contrast to stability, can be assumed from a decolonial perspective. This is perhaps why Malabou struggles with the concept of "destructive plasticity" [14], which, by identifying a form of destruction, assumes a stable vantage point from which this judgment is obtained. Instead, evidencing the logic of epigenesis, decolonial critique remains at the impact point, working within localized epicenters where tremors are most

concentrated, inhabiting cracks and ruptures as alternative fields of enunciation that are not undergirded by delineations of stability-instability (as judgments obtained from "on high"). This possibility, in my mind, is what is at stake in the "whence" of materialism that I propose.

### 4.4. Malabou with and beyond Dussel

Decolonial critique, in Malabou's sense, neither speaks in a language nor seeks the possibility of legitimate enunciation in the European "mother tongue". To this end, she notes that Dussel's decoloniality pursues a "pluriversal transmodern world"—or "an era that comes after postmodernity, a kind of post-postmodernity, not restricted to Europe and open to the whole world" [15] (p. 183). As a universal project of diversity in which critical epistemic interventions would be issued from the margins, Dussel's decolonial concept of transmodern philosophy operates "in translation", rather than "in the mother tongue" [15] (p. 185). Yet, I find in Malabou an implicit critique of Dussel's "critical decolonial thinking" insofar as it assumes that across geopolitically situated knowledges, "a real horizontal dialogue and communication could exist between all peoples of the world (Grosfoguel 2011: 28)" [15] (p. 183) [15]. This is to say, Dussel's "transmodernity" seems to assume that subaltern knowledges, no longer marginalized within a post-postmodern era, could traverse different geopolitical locations "without encountering major difficulties of comprehension or of translation between incompatible thinking paradigms" [15] (pp. 182–183). Building on but going beyond Malabou, I suggest that Dussel (as well as deconstructive thinkers in their reliance of a pernicious form of a "mother tongue") maintain an investment in the possibility of dialogue and, hence, continue to presuppose intelligibility as an antecedent, even if "anonymous", condition of shared life. In this way, they remain attached to the tradition of Western philosophy, in particular its concept of "logos" as precisely such an anonymous pre-condition that, while not itself intelligible, nonetheless constitutes a universal horizon of intelligibility.

This assumed "universality of spirit" [15] (p. 185) takes shape in relation to language (i.e., symbolic representation, or meaningfulness as it is based on tradition) [16] as a defining feature of "human" life and a necessary condition for sociality and politics. As Malabou elsewhere notes, in this respect, "contemporary philosophy bears the marks of *a primacy of symbolic life over biological life that has been neither criticised nor deconstructed*" [20] (p. 228). I find this kind of attachment to language to be linked with antecedent intelligibility, as evidenced in the teleocratic modern nation state. As a democratic form of "human" politics, the model of the state positions dialogue as a requisite basis of social–political life: the efficacy of which, from a modern perspective, is constituted by appeal to a universal rationality. Even when this linguistic basis is critically approached in its situated socio-cultural contingency, and not as an autonomous discourse of truth (as deconstruction does, for instance), Language remains the "anonymous" ocean upon which philosophy and politics drift. This is to say, while language may constitute an outside of truth, there is nothing outside of the symbolic.

As Eduardo Kohn argues, this commitment to language reflects a homogenizing conflation of representation with language (i.e., the symbolic), as well as an anthropocentric assumption that semiotics is a uniquely human activity occurring only based on established human socio-cultural contexts [19] (p. 8). Building on Malabou's engagement with Grosfoguel, I suggest that this attachment is more than a theoretical commitment to the "filter" of European philosophy or to a democratic model of politics [15] (p. 184). It is also, in my view, an affective attachment to the antecedent legitimacy of *ego conquiro* and, hence, to "'the political, economic, cultural, and social conditions of possibility for a subject who assumes the arrogance of speaking as though it were the eye of God... ([Grosfoguel] 2012: 89)" [15] (p. 181). This uncritical attachment to coloniality (as manifest in the uncritical primacy of the symbolic) can occur for philosophers of the "center" who, as such, find themselves beneficiaries of an antecedent geopolitical legitimation.

As I believe the above critique of Dussel suggests, an attachment to coloniality can also occur for critical philosophers of the periphery who, finding themselves always already on the outside, gaze within, longingly. In this respect (and in anticipation of my discussion of María Lugones' decolonial feminism), I note Lugones' critique of Walter Mignolo, for whom decolonial "enunciation is enacted from the subaltern perspective as a response to the hegemonic discourse and perspective (Mignolo 2000, x)" [21] (p. 752) [17]. Although, for Lugones, dialogue is both possible and "necessary for those resisting dehumanization in different and intermingled locales" [21] (pp. 752–753), she does not, like Mignolo, see this as occurring toward the possibility of mutual understanding with "the modern man". She does not assume a universal horizon of intelligibility or remain attached to language as symbolic representation. Instead, taking the form of what Lugones calls "complex communication" [23] (which, in my view, engages non-symbolic and more-than-human semiotic modalities and, moreover, is resonant with my discussion of a language of names below), "dialogue" materially transpires "toward a newness of be-ing" [21] (p. 753) [18], toward possibilities of transdifferentiation.

*4.5. Language of Names*

Resisting the "'theoretical imperialism'([Marchant] 1987: 309)" [15] (p. 185) implied by an assumed universality of spirit and an antecedent "mother tongue", Malabou turns to Patricio Marchant, who doubts the possibility of translation in Latin America. This is because, in colonial contexts, translation continues to subordinate non-Western languages and traditions to Eurocentric/colonial criteria of legitimacy as well as to the necessity of the symbolic as the medium of (human) relation. Malabou notes that for Marchant, the struggle between a "future Hispano-American language" and European Spanish does not occur with respect to possibilities of translation but instead pertains to "the need to be, to be *a name, reparation for the violation as a 'new' language* (1987: 317)" [15] (p. 185). In this, Malabou suggests that decolonial critique takes the form of a non-symbolic and poetic "language of names", rather than "a Language reduced to 'words' (means of communication)" [15] (p. 185).

Against the ideal of trans-geopolitical dialogue, a language of names calls for the crystallization of untranslatable and "irreducible singularities" [15] (p. 185) from out of the situated tensions of colonial contexts shaped by legacies of asymmetrical encounters between incommensurable traditions of thought. As such, decolonial critique "precisely contradicts the anonymity of language", or the deconstructive "outside" [15] (p. 185). Decoloniality, in this sense, neither functionally constitutes an effacement of the Cartesian subject nor "opens" a subaltern site of dispossession (as deconstruction does). But it also does not pursue linguistic possibilities of translation based on the assumption of global symbolic intelligibility (as suggested by Dussel) or seek the preservation of a non-Western tradition.

Instead, through inhabitations of subaltern loci of illegitimacy, a "critical" decolonial enunciation is constituted by a poetic mixture of heritages—for instance, between "indigenous and Christian values" (as seen in Neruda's *Alturas de Macchu Picchu*) [15] (p. 185)—that, without reconciliation or attachment, "points to the paradox of identity in Latin America" [15] (p. 186). Arising from the "surface" tensions of localized epicenters, Latin-American poetry is a mode of critique that materializes a "seismic" attunement to the social as a dynamic and heterogeneous field. It is not about writing as a symbolic practice of "leaving and then interpreting traces" (where "traces are always traces of something" intelligible for someone and where "someone" ends up aligning with the antecedently legitimated subject positionalities described above) [8] (p. 111). It rather indicates a mode of writing that is perhaps better understood under the "scheme" of plasticity in the sense that poetic "traces" would not be "signs of something else, but forms-in-formation, including transformation and annihilation of form itself" [8] (p. 111). In this respect, I point out that Malabou's turn to a language of names, which is also a move away from modern literature, could be seen as paralleling her "evolution" of deconstruction, which "shifts Derrida's core emphasis on 'arche-writing,' graphic inscription, and the 'trace'" to plasticity: "to the



question of *form* and the manner in which forms survive deconstruction and persist or are transformed in excess of any logic or possibility of self-identity or presence" [2] (p. 3).

Decolonial critique is likewise about non-symbolic possibilities of representation as these occur anarchically. It is about creating resistantly in/with/through social–political upheavals without the support of antecedent legitimacy, discursive embeddedness in a tradition, or appeal to the harmonizing force of universal spirit. From out of the nothingness of subaltern loci, decolonial enunciation manifests a transdifferentiating mode of agency that "changes difference", epigenetically giving rise to new and subversive possibilities of being. The task, as I argued above, is therefore to "see" this anarchic and decolonial mode of critique/resistance as it is already occurring within postcolonial contexts, and not, as Malabou's "whither" of materialism suggests, to make it possible.

### 5. Lugones' Decolonial Feminism

Just as decolonial thinkers lead Malabou to turn away from philosophy as a global theoretical field unified by the symbolic, so too do they shed light on the heterogeneity of colonial–modern social–political conditions, wherein divergent social forms and logics collide without reconciliation. In this respect, I turn to Lugones' decolonial feminism, which comes as a critical development of the decolonial theorists discussed by Malabou, to elaborate the social–political implications of Malabou's delimitation of the decolonial outside [19]. Resonant with Malabou's critical development of deconstruction, Lugones is not interested in modes of critique or resistance that presuppose the kind of antecedent legitimation and stability characteristic of deconstructive critique in view of its enunciative legitimacy. She does not, then, pursue resistance as an immanent destabilization of coloniality–modernity that occurs through critical inhabitations of dominant social positionalities of privilege. Rather, she is interested in immanent resistance to colonial domination as it collectively arises from everyday inhabitations of what Malabou would regard as sites of "definitive exile" or from the "political voids" of colonial contexts.

Reminiscent of Malabou's brief discussion of the decolonial concept of "borderlands" [15] (p. 182) and her related recognition that, as Grosfoguel argues, "decolonial thinking inhabits 'exterior spaces not fully colonized by the European modernity' but still coexisting with the colonised ones (2011: 28)" [15] (p. 182), Lugones analyzes subaltern "fractured" loci [21] (pp. 752–753). For Lugones, subaltern loci of enunciation and agency are fractured at the "colonial difference"; which, drawing from Mignolo, she regards as a "physical and imaginary location of confrontation of two kinds of local histories (Mignolo ix)" [21] (p. 752). In this, she understands subaltern loci as co-constituted by colonial domination—or by what Malabou regards as the "definitive exile" of the colonized—*and* by non-dominant forms of resistance to colonial domination, as these are informed by other local histories: those of the colonized. Going beyond Mignolo's use of the concept in relation to her above-discussed critique, Lugones thus finds that there are "many colonial differences, but one logic of oppression" [21] (p. 755). Based on this, I shift away from a critical perspective that looks to "the" political void toward a decolonial feminist perspective that engages political voids as divergent sites of enunciation and agency that reflect the heterogeneity of postcolonial topographies.

To clarify the specificity of subaltern fractured loci, as well as to show the irreconcilability of the local histories at play in their divergent constitutions, Lugones draws from Juan Ricardo Aparicio and Mario Blaser to distinguish between "modern" and "non-modern" logics [21] (p. 742). The former is a "categorial, dichotomous, hierarchical" logic of domination and is central to the colonial–modern nation state and its social formations, namely, "to modern, colonial, capitalist thinking about race, gender, and sexuality" [21] (p. 742). I note that, as such, modern logic can also be seen as teleocratic. Logically "at odds" with modern social logic—and, hence, untranslatable in modern terms—Lugones sees non-modern social logics as encompassing divergent Indigenous "knowledges, relations, and values, and ecological, economic, and spiritual practices" [21] (p. 743). As such, non-modern social logics do not constitute the "exteriority" of modernity [21] (p. 749); they are not

primitive ways of life from a "pre-modern" past [21] (p. 743). Rather, Lugones maintains that non-modern social logics play an ongoing (even if at times difficult to trace) role in the resistant organization of the social [21] (p. 749).

In this way, Lugones sheds light on the social as a radically heterogeneous, indeterminate, dynamic, and plastic field of agency that is shaped by tensions between incommensurable social logics. To evoke again Malabou's geological imagery, for Lugones, the social is perhaps like the crust of the Earth: a shifting field (or habitat) marked by fault lines, by cracks where social forms and logics, like tectonic plates, contingently meet. Grinding together, co-extensive socialities accumulate tensions that unpredictably erupt, constituting localized epicenters of social upheaval. Lugones, in this respect, can be seen as offering an anarchic approach to sociality, one that assumes no antecedent foundations for social formations (e.g., "gender" or symbolic language) or, for that matter, emergent agencies. Moreover, she makes clear the porosity and permeability of the social, in the sense that (as James puts it with respect to Malabou's key insights), "no system, no form, no regime of being is ever closed or self-sufficient. All are constitutively open to and affected by other systems, forms or regimes to which they are necessarily related and are so in such a way as to ensure that they will always have a capacity to modify or transform themselves" [2] (p. 4). This brings social–political exteriority into focus as a situated "outside" that is constituted by enfleshed social–cultural tensions and, through embeddedness in colonial legacies, split at the colonial difference. It is in relation to this sense of, as Alejandro Vallega might say, "radical exteriority" [18] (pp. 68–75), that Lugones approaches subaltern fractured loci as presently inhabited sites of immanent resistance to colonial domination.

Lugones's decolonial feminism critically builds on, and indeed radicalizes, the work of decolonial thinkers in order to pursue what I view as a plastic and anarchic approach to sociality and resistance [20]. For instance, her "coloniality of gender" (which is a precursor to her decolonial feminism) shows that by presupposing colonial–modern determinations of "sex" and "gender", Quijano's "coloniality of power" assumes a heterosexual matrix and remains tied up with a distinctive form of social homogeneity and domination [29] (pp. 17–18). In other words, in relation to anarchism, Lugones can be seen as critiquing Quijano insofar as he preserves colonial–modern constructions of 'human" sex and gender as the *arkhé* and *telos* of the social. As she argues, such an implicit preservation of social homogeneity erases the colonial difference and imposes on the oppressed "an order of relations uncritically as if coloniality had been completely successful both in erasing other meanings and in people having totally assimilated" [30] (p. 31). Decolonial theories such as Quijano's therefore render the oppressed "nameless" (to follow Malabou's language), concealing them "as the people they have been, are, and becoming in a line of continuity woven by resistance to multiple forms of coloniality" [30] (p. 36) [21].

But Lugones does not, in this respect, see the anonymity of the oppressed only as a consequence of domination. For her, the colonized are also nameless insofar as, backed by incommensurable non-modern social logics and senses of community, their inexhaustability remains invisible to perspectives that "see" only on the basis of modern teleocratic logics [22]. She writes: "the meaning of the resistance will be unintelligible to the oppressor and may be done with or without critical reflection, but always without an understanding in common between oppressor and oppressed" [30] (p. 34). Lugones, in other words, does not presuppose universal intelligibility where social agency is concerned. Indeed, she suggests that to do so would be to participate in the erasure of the colonial difference, which, for her, is constitutive of coloniality [21] (p. 749).

Beyond an uncritical (and colonial) assumption of social homogeneity, Lugones thus finds it necessary to learn to "see" the colonial difference without positioning mutual understanding as an ideal or seeking to render opposites commensurable through, for instance, determinations of sameness and difference [23]. For her, this occurs through a decolonial methodology. Through a non-imperialistic sensitivity to divergence (in particular as informed by non-modern logics of complementarity and reciprocity), a decolonial methodology:

> . . . reveals more than one reality. The reality of the dominators who imagine the peoples to be animals, beasts, dangerous cannibals and aggressive sexual beings; the realities of those who see and resist the coloniality within it and are as resistors constituted by the cultural, relational, cosmological shared practices, values, knowledges that animate their resistance [30] (p. 28).

Sustaining such a "double vision" at/of the colonial difference, Lugones' decolonial feminism allows for a non-exhaustive approach to colonial domination, wherein "one sees both the reduction and the resistance of people instead of dependent nation-states" [30] (p. 28). In this respect, and like Malabou and non-classical anarchists, Lugones does not approach domination and resistance as distinct forces but instead views power as a fractured locus of tension constituted by an ongoing "oppressing $\to \leftarrow$ resisting" relation [21] (p. 746).

Lugones too, then, pursues resistance not as external opposition to domination but as an anarchic, generative–destructive force that, in her case, arises immanently, epigenetically from the "cracks" of colonial–modern social formations: or, in other words, from fractured subaltern "political voids" that are presently inhabited by the oppressed at the colonial difference. Yet, by seeing the social as already heterogeneous—or as already plastic—Lugones' decolonial feminism does not assume as Malabou does (especially under a deconstructive guise) that anarchism needs to be made possible or that political voids must be opened—for this, she suggests, would only be to buy into coloniality. The task of decolonial feminism is not, then, a "whither" but a "whence": a learning to "see" the radical social heterogeneity and plasticity of colonial contexts, a learning to "see" subaltern loci embedded in these conditions as divergently "fractured both by logical difference and by resistant presence" at the colonial difference [21] (p. 749), and a learning to "see" immanent resistance to colonial domination, as this is informed by the memorial traces of nonmodern social logics and already arising within the everyday lives of the oppressed.

## 6. Conclusions: Decolonial Plasticity

Looking to subaltern loci as openings to deeply relational and anarchic social fields of agency fractured at the colonial difference, for Lugones, "the sole possibility of such a being lies in full inhabitation of this fracture, of this wound, where sense is contradictory and from such contradiction new sense is made anew" [21] (p. 752). This resistant sense-making, she argues, occurs communally through the multiple perception of "a being who begins to inhabit a fractured locus constructed doubly, who perceives doubly, relates doubly, where the 'sides' of the locus are in tension, and the conflict itself actively informs the subjectivity of the colonized self in multiple relation" [21] (p. 748) [24]. I see in such lived inhabitations of social heterogeneity an anarchic and transdifferentiating "power to style" that reflects the seismic logic of epigenesis. This entails sensuous attunement to the equilibrating and non-localizable affective dynamics of fractured loci—one that, as indeterminate tensions play out in/with/through the body, creatively "adapts to its negotiation always concretely, from within, as it were" [21] (p. 753) [25]. Furthermore, insofar as everyday resistant inhabitations of fractured loci are, for Lugones, "grounded in a peopled memory" [21] (p. 754), I point out that seismic agency also involves a transindividual "genealogical" [26] attunement to enfleshed, non-modern, and "anachronic" [27] lineages through which "the colonized have kept a sense of self struggling against dehumanization, against assimilation, keeping resistant senses of self, transculturated, recovered, or new" [30] (p. 40) [28]. For Lugones, such an affective, genealogical sensibility is key to nondominant forms of resistance.

As an attunement to the corporeal traces of divergent social logics, this genealogical sensing resembles the kind of remembering "the past without having necessarily memorised it" [24] (p. 27) that is taken up by Malabou in her recent re-elaboration of anarchism. Here, she turns to Peter Kropotkin to approach contemporary forms of mutual aid (for instance, arising in response to COVID-19) as an immanent biological resistance to biopolitical domination [24] (p. 27) that "proceeds from a memory, the non-conscious remembrance of past connections that allow for future possible social networks" [24] (p. 28). I draw from

this to gesture to the fact that, especially when brought together, Lugones and Malabou shed light on resistant agencies, as these arise from a more-than-human, affective, and genealogical horizon of sociality that emerges precisely at the vanishing point of dominant metaphysical/ontological determinations: for instance, colonial–modern racialized constructions of the "human" and "nonhuman" (Lugones), or biopolitical designations of biological and symbolic life forms [24] (p. 27) [29].

To this end, I look to Malabou's "new materialism"—which "asserts the coincidence of the symbolic and the biological" and, consequently, maintains that "there is but one life, one life only" [20] (p. 235)—to suggest that plasticity (as referring to a "space" of corporeal indeterminacy and porosity between biology and the symbolic) constitutes a permeable, more-than-human order of relationality. I take this to be an affectively mediated sociality that antecedes individuation and transpires "without chains" [24] (p. 19). This echoes Lugones' engagements with non-modern relationalities and cosmologies, as well as Peter Kropotkin's theory of "mutual aid", which, as discussed by Malabou, not only gestures to trans-species possibilities of cooperation but more fundamentally to "a contingent dynamic relation between living beings, *always anterior to individuals,* that preserves individual variations" [24] (p. 26, my emphasis). I see this prior affective relationality as the indeterminate and non-localizable basis of transdifferentiating and epigenetic social agencies—as an anarchic and indifferent movement that, like an earthquake, can unpredictably give rise to destructive transformations of the social within orders of domination and explode modern determinations of the "human". To this end, I view political phenomena (such as immanent resistance to domination) not as "cultural" or "historical" but rather as plastic in that they erupt at/as the vanishing point of biology–symbolic, nature–culture, human–non-human.

Based on genealogical attunements that traverse this corporealized field of indeterminacy—with, as Omar Rivera might say, the radical indifference of "border sensibilities" [36] (p. 245)—Lugones finds that "she awakes in her embodied self a double feeling/consciousness of the permeable body, and then [moves to] to discover, explore, appreciate, engage her permeable body in reciprocity, in an unstable, dynamic balancing" [26] (p. 277). Like Malabou on plasticity, it is through such embodied and non-localizable balancing acts that, for Lugones, new ways of being can materially arise. Drawing from Mary Louise Pratt, Lugones refers to this sensuous, generative–destructive movement as "transculturation" [30] (p. 34). Resonant with Marchants' language of names, transculturation concerns "the multifaceted process in which hegemonic cultures influence subjugated ones, in which subjugated cultures give up old and acquire new values and meanings, in which completely new cultural forms are created (See Horswell 2006, 6)" [30] (p. 35).

As a plastic and anarchic process of formation that proceeds with indifference to both possibilities of critique and preservation of a tradition, transculturation "always brings in the shared culture, ways of life, ways of knowing, understanding the self in relation that are not static, rather they are always changing and transforming the meanings of colonial, modern, capitalist structures of meaning (See Ortiz 1995, Pratt 2007)" [30] (p. 35). In this respect, transculturation can be understood in relation to the challenge of epigenesis: "the possibility of thinking historical transformation from within, without recourse to either telos or ground, but rather with attention to contingency, emergence, and movements of propagation from one element or domain to the next" [7] (115).

Further resonant with epigenesis, specifically as a way of thinking that does not presume human exceptionalism [7] (p. 109), I note that in Lugones' elaboration of it, transculturation is not reducible to a "human" activity. This is apparent in that, informed by traces of non-modern social logics, transculturation is constituted by ways of life that, from a colonial–modern perspective, have antecedently been constructed as "non-human" or, as discussed above, have been positioned as a "premodern" outside of "culture" [30]. Moreover, Lugones's turn to "decolonial aesthesis", or "the decolonial deconstruction of aesthetics that privileges the senses" [26] (p. 275), suggests that transculturation (as well as a decolonial language of names, perhaps) is based in "the body and permeability and all that permeability allows us to reconceive about the world we live in" [26] (p. 275) [31]. In other

words, it is a sensuous movement that transpires affectively, in excess of a "human" body (as the antecedent site of spectatorship in Western aesthetics), as well as symbolic language.

Arguing that social formation needs to be understood "through accessing the process of creation, the process of transculturation, rather than taking at face value the organization of the social" [30] (p. 35), Lugones too, pursues a non-intentional mode of creative agency that, like Malabou's transdifferentiation, is immanent to plastic and anarchic processes of social–cultural (trans)formation and cannot be understood through appeal to dominant modern logics or conceptual determinations. Lugones refers to this minimal form of resistant agency as "active subjectivity" [21] (p. 746) [32] and sees this as including:

> habit, reflection, desire, the use of daily practices, languages, ritual knowledge, a thinking-feeling way of decision making, which may not be part of the meanings of the institutional and structural meanings of the society but may be part of the meanings in the resistant circle [30] (p. 34).

For her, it is through active subjectivity that inhabitations of fractured loci enact transculturation as "a critique of racialized, colonial, and capitalist heterosexualist gender oppression as a lived transformation of the social" [21] (p. 746) and, hence, sustain resistant ways of living in the midst of colonial violence [26] (p. 277).

This, I suggest, is not unlike the view that local anarchist governance and mutual aid groups are "signs" (i.e., forms-in-formation) of a non-hegemonic form of solidarity and a power "that will allow us to re-discover unlived potentials and avoid an unliveable future" [12] (p. 120). In this respect, I point out the significance of non-modern socialities for anarchism—as noted, for instance, by Conty, who acknowledges that "small, local, anarchist groups are learning from indigenous peoples that never developed pyramidal hierarchies how to protect the good life, the *buen vivir* and how to cherish the hope that a better future lies waiting in the unlived potentials we all harbour" [12] (p. 120). As Lugones makes clear, active subjectivity (as an anarchic, seismic power of resistance) is thus not the end goal of political struggle. It is, rather, its beginning and possibility—a "whence" that, as a radical "coalitional starting point [21] (p. 753), "affirms a profound term that Maldonado Torres has called the 'decolonial turn'" [21] (p. 755).

**Funding:** This research received no external funding.

**Informed Consent Statement:** Not applicable.

**Data Availability Statement:** No new data were created or analyzed in this study. Data sharing is not applicable to this article.

**Conflicts of Interest:** The author declares no conflict of interest.

## Notes

1    I draw here from Kelli Zaytoun's use of the term "process" metaphysics, which, in distinction from substance metaphysics, is used to characterize the focus of Nahua philosophy [6] (pp. 20, 28). I do so to acknowledge the underexplored pertinence of indigenous knowledge systems to new materialisms such as Malabou's (see [6] pp. 25–27). For more on Malabou and "process philosophy", see [7] (pp. 141–142), as well as the risks of (p. 144).

2    To be clear, this is not, as Tyler Williams cautions against, to position destructive plasticity as a tool for resistance [1] (p. 6).

3    Plasticity is not elasticity, namely, the biological capacity to return to an original form after undergoing deformation [3] (p. 15), nor is it flexibility, the capacity for endless polymorphism [3] (p. 12).

4    For Malabou, plasticity and epigenetics are "regarded as identical or synonymous" [7] (p. 133).

5    For instance, as a "biological phenomenon, a transcendental structure, a science (epigenetics), and a hermeneutical instrument" [7] (p. 120). See [8] (p. 118) and [9] for more on Malabou's understanding of epigenesis.

6    See, for instance, [1] and [10].

7    For Malabou, the logic of epigenesis does not presuppose a stable, reflective subject position. This is clear in that the kind of perspectival stability needed to conceptualize an eruption in reference to its "hypocenter" is only possible "after and without this changing anything that has occurred [7] (p. 121). I note resonance between the logic of epigenesis and Lugones' discussion of "tactical strategies" [11] (pp. 207–237).

8    In this way, Conty could be seen as responding to Malabou's claim (following Althusser) that "the same plasticity as the bio-logical one should prevail in the social and political order" [4] (p. 208).

9    For more by Malabou on anarchy, in particular concerning the relationship between philosophy and anarchism, see [14].

10   To this end, see María Lugones' critique of Marx, who, she argues, assumes the social exhaustiveness of capitalist oppression and thus positions the laborer as thoroughly alienated from their own agency, including its resistant exercises [11] (pp. 53–54). It follows from this that resistance (in the form of revolution) must be made possible through workplace organization against capital. Yet, she points out, insofar as the logic of capitalism is presupposed as a totalizing mode of domination, it is unclear how such a class struggle is possible in the first place. In this respect, Lugones finds Marx's appeal to the necessities of unstable structures as the source of class consciousness to be insufficient. See also Malabou's critique of Marx's materialism, which sim-ilarly problematizes his grounding of revolution in a dialectical logic of teleology and necessity [4] (p. 213).

11   See also William's discussion of Malabou's proposed "task" [1] (pp. 12–14).

12   See also [16].

13   As Omar Rivera explains with respect to Linda Martín Alcoff's reading of Dussel, from the perspectives of the subaltern, "the linkages between power and knowledge tend to be acknowledged, not in order to reject the possibility of justifiable knowledge, but to understand political contexts as factors belonging to normative epistemological determinations. This would constitute a 'political epistemology'" [17] (p. 44).

14   See [2] (p. 11).

15   See Alejandro Vallega's similar critique of Dussel [18] (pp. 88–92) and Rivera's analysis of this critique [17] (pp. 50–54).

16   I draw here from Eduardo Kohn's discussion of symbolic representation: "signs that are conventional, systemically related to one another, and "arbitrarily" related to their objects of reference" [19] (p. 8, see also 32).

17   I note similarities between this and Silvia Rivera Cusicanqui's critique of Mignolo. See [22].

18   As Kohn's "more-than-human" anthropology makes clear, a critique of language (i.e., the symbolic) need not entail a rejection of semiotics. In this respect, drawing from Peircean semiotics, Kohn approaches the Amazon rainforest as a dynamic relational ecology constituted by non-symbolic modes of representation (e.g., iconic and indexical), which, for him, are shared by all bi-ological life [19] (p. 8). This non-anthropocentric approach to representation, I find, is helpful in understanding the critique of language that is a part of not only of recent developments in anarchist thinking (see Malabou [24] (pp. 21–26), as well as ac-counts of biosemiotics [25] (pp. 133–135), but also the aesthetic turn in decolonial philosophy taken by Lugones, Rivera, and Vallega, among others (see Lugones [26] (p. 275) and Rivera's critique of Dussel [27] (pp. 161–178)). Moreover, as Zaytoun suggests with respect to Gloria Anzaldúa's "poet-shaman aesthetics" (developed in relation to Indigenous philosophies) [6] (p. 27), I point out that rejection of the symbolic need not involve a turn from language. It can entail, for instance, engagement with words not as symbols but as material forces (perhaps, as Kohn would say, as iconic or indexical "sonic images" [19] (p. 30)) that, as such, play "an integral role in the composition and transformation of reality" [6] (p. 26). In this respect, Zaytoun notes that for Anzaldúa, writing is not reducible to linguistic communication because it also serves as a mode of participation in/with the weaving movements of the cosmos [6] (p. 106).

19   I note that while the turn from "whither" to "whence" resonates with Malabou's engagement with Kropotkin's anarchist theory of mutual aid (which will be discussed in the conclusion), I choose to elaborate this anarchist turn in relation to decolonial theory, specifically Lugones' decolonial feminism. In my view, this is important not only to pursue the productive theoretical generations that can occur when anarchist theory and decolonial theory "collide". It also sheds light on the significance of decolonial theorization, in particular of exteriority and critique, in the context of Malabou's larger body of work.

20   For more on a new wave of decolonial philosophers who pursue what I see as an anarchic direction, see [28].

21   I note that Lugones has a similar critique of Foucault [30] (p. 34).

22   This is not unlike Malabou's view that, with respect to biological resistance to biopolitics, "the articulation of political discourse on bodies is always partial" [20] (235). See also her political account of the "anonymous masses" [24] (p. 27), which I find is resonant with my discussion of the namelessness of the colonized, as well as my below discussion of resistance.

23   See also Rivera's critique of Quijano [31] (pp. 141–171).

24   I note resonance with Miller's account of the "subject" of epigenesis: "This subject, newly aware, is certainly not to be under-stood as individual: Neither homunculus (with its reiteration in the 'selfish gene') nor 'self' (the individual subject of free will), the subject of epigenesis takes shape as a process of transcendental—and transindividual—becoming" [7] (p. 117).

25   See also Lugones' interpretation of Anzaldúa's "germinative stasis" [32].

26   I draw here from Nancy Tuana and Charles Scott's development of "genealogical sensibilities", in particular with respect to Anzaldúa's concept of "nepantla" [33] (pp.108–142). See also Suzanne Bost's account of how such a "backward looking" figures in the work of Anzaldúa, in particular as this is drawn from indigenous traditions and pertains to an "oth-er-than-humanist ethics" [34] (p. 1569).

27   See Vallega's concept of "anachrony" [18] (pp. 115–119).

28   See Cynthia Pacccacerqua for an account of an affective inhabitation of an "immemorial reality" without personal history [35] (p. 342).

29    See also Gearóid Brinn and Georgina Butterfield's discussion of "realist anarchism", which draws from Malabou [25] (p. 132) to pursue a trans-species sociality and non-anthropocentric account of political organization [25] (p. 135) that reflects "a realistic, materialist understanding of existence beyond the human species" [25] (p. 127).

30    Lugones points out how the modern epistemological apparatus actively reduces the non-modern social logics of the oppressed to "premodern ways" [21] (p. 743). In this, "modernity attempts to control, by denying their existence, the challenge of the ex-istence of other worlds with different ontological presuppositions. It denies their existence by robbing them of validity and of co-evalness" [21] (p. 749).

31    I note that plasticity could also provide a way of pursuing permeability or embodiment beyond a Western, dualistic framing. For example, Malabou writes: "one pertinent way of envisaging the "mind–body problem" consists of taking into account the di-alectical tension that at once binds and opposes naturalness and intentionality and in taking an interest in them as inhabiting the living core of a complex reality. Plasticity, rethought philosophically, could be the name of this entre-deux" [3] (p. 82).

32    For more on active subjectivity, see [11].

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
