# Peer review of "From “Whither” to “Whence”: A Decolonial Reading of Malabou"

_philosophies, doi:10.3390/philosophies8050093_

Round 1
Reviewer 1 Report
This article is engaging, articulate, well researched, strongly argued, and compelling. This reviewer recommends it for publication. Without challenging any of the overall argumentation, which the author presents in a coherent and straight-forward manner, the following remarks simply serve as opportunities for brief consideration should the author and/or editors be so inclined.
· Stylistically, the transition from Malabou to Lugones between sections 4 and 5 of the essay appears quite sharp. Some softer transition that introduces exactly what moves the author from Malabou to Lugones would be helpful.
· Section 1 of this article seems a bit too brief, in this reviewer’s opinion. Perhaps a synthesizing sentence after the first two sentences would help the author articulate the precise intervention the article is making. The last sentence of section 1 is not exactly clear what’s at stake in the article. At line 25, the term “social-political plasticity” arrives without any explanation.
· The last paragraph of section 2 does not explain how the “explosiveness” of plasticity describes a destructibility already embedded within plasticity’s (versus elasticity’s) metamorphic potential. For that reason, the emphasis on explosion might not be clear to readers less acquainted with Malabou’s work.
· Between sections 2 and 3, the author’s description of “transdifferentiation” seems to parallel Malabou’s descriptions (in Before Tomorrow, Plasticity: The Promise of Explosion, and elsewhere) of the epigenetic over the teleologically genetic. Attention to the epigenetic here (for example within lines 57-64) would help account later (for example, between lines 89 and 107) for the materialist senses of resistance Malabou invests in the “regenerative” and the “ateleological.”
· In “Philosophy and the Outside,” part of Malabou’s point is that the figure of the outside remains consecrated by the traces, heredity, and authority of the inside. The need to think the outside is thus the need to think what Malabou calls the outside of the outside. This point does not challenge anything about the author’s argument, but this formulation goes unaddressed in section 4 of the article and might be of service to the author’s argument.
· Because Derrida famously stresses that deconstruction is always already at work in the work, is already an internally destabilizing force (an excentric center) within Western metaphysics, it is not clear exactly how controversial the author’s claim is in section 4.1 that deconstruction cannot occupy this position of an anarchic antecedence. Of course, deconstruction articulates the quasi-transcendental necessity of this trace-structure, but it does not claim to be able to bring this exteriority to presence. For this reason, as Malabou and the author point out, deconstruction is never absolutely removed from a tradition of philosophical and philosophy’s colonialism. But it is nonetheless important, this reviewer thinks, that Malabou stresses from the first line of her essay, that she is foremost interested in the confrontation between these two forms of the outside. This logic of confrontation (which for Malabou is itself ateological – as evidenced by her argument in her essay “Whither Materialism”) does not get addressed explicitly by the author and might be worthy of consideration.
· Section 4.3 elides the role “literature” plays in the argumentation. It would be interesting to hear what the author thinks about this connection between the references to naming and poetry via Marchant and Neruda versus the earlier treatments of “modern literature” as “outside” via Foucault. Admittedly, this is a detour of the topic at hand and this reviewer is aware that the author might not—and quite clear does not—want to take the article in this direction.
· Lastly, the article might benefit from more engagement with the secondary literature devoted to Malabou’s work. There is none in the article’s bibliography or notes. The author might consider Burdman’s reading of contingency and epigenesis (2016), Crockett’s account of Malabou’s situation between biology and deconstruction (2018), Williams’s description of Malabou’s materialism as an anarchic ontology (2019), the multi-authored collection on epigenesis in October (2021), or Ian James’s description of plasticity as both a material mutability as well as a conceptual/discursive passage (2022), to name just a few.
Reviewer 2 Report
This is a well written and organised engagement with Malabou's work that brings her into dialogue with a significant postcolonial thinker. As such it builds on her existing engagement with postcolonial thinkers and draws insightful parallels, while at the same time advancing the author's own arguments and interpretations. I am happy to recommend it for publication, and only have the following suggestions for minor changes:
1. The article would generally benefit from more signposting. The author introduces several significant concepts and theoretical traditions (plasticity, deconstruction, teleocracy, etc.). These are clear enough, but it is not always clear to the reader why these ideas are being introduced and why they are necessary for the overall purpose of the article.
2. The notion of shifting from the 'whither' to the 'whence' of materialism is clear and well articulated, but it is sometimes unclear whether this is represented as a shift that Malabou undertakes herself, or whether it is one that the author is recommending. It is suggested that this view is "based on" or "implicit in" Malabou's writings, but the formulation at the end of Chapter 4 seems to imply that this is a turn Malabou herself has not made (at least consciously). Perhaps a few sentences could be added clarifying how the author sees this shift.
3. The main focus of the article is Malabou's writing on Latin American decolonial theory, but the conclusion also mentions her engagement with Kropotkin, which might also be read as particularly concerned with the 'whence'. Given this, perhaps this connection could be made earlier as part of the discussion of the shift from 'whither' to 'whence', and the choice to focus on her engagement with decolonial theory rather than her work on Kropotkin and anarchism contextualised and explained.
The quality of the English is very high, with the only real errors limited to typos of the sort that are to be expected in any text (though please note that Malabou's name is misspelled several times in the list of references). One (very minor) issue of style is that the author tends to use first person statements in the simple present that sound more like a summary or abstract than the main text (e.g. "I find", "I point out"). These are a little jarring, and could be replaced with other, less direct, formulations (or in some cases turned into declarative sentences without the first person construction).
